# ILPG: INSTANCE-LEVEL PROTOTYPE GENERATION FOR ZERO-SHOT LEARNING

## ABSTRACT

*Zero-shot learning* (ZSL) aims to recognize unseen classes by transferring knowledge from seen ones through shared semantic attributes. However, existing methods typically align image features with static, class-level prototypes, which ignore intra-class diversity, lack adaptivity to individual samples, and often suffer from semantic drift. We propose the *Instance-Level Prototype Generation* (ILPG) network, a lightweight framework that dynamically refines semantic prototypes on a per-instance basis. ILPG combines an attention-based attribute localization module, which highlights discriminative visual regions, with a semantic adjustment pathway that personalizes class prototypes to capture instance-specific variations. This design achieves fine-grained alignment between image features and class semantics while mitigating prototype rigidity. To further enhance robustness, we introduce a synergistic loss formulation that balances discriminability and semantic consistency, ensuring dynamically adjusted prototypes remain semantically faithful. Extensive experiments on three widely used benchmarks (CUB, SUN, and AWA2) demonstrate that ILPG consistently outperforms competitive baselines. ILPG not only establishes new state-of-the-art performance in both conventional and generalized ZSL but also provides interpretable attribute–feature associations.

## 1 INTRODUCTION

Zero-shot learning (ZSL) has attracted increasing attention in recent years. The goal of ZSL is to correctly recognize samples from unseen classes during testing, even though no labeled instances from these classes are available in training. This setting is highly relevant in practice, for example recognizing rare species in biodiversity monitoring, detecting uncommon conditions in medical imaging, or handling long-tail classes in open-world environments. Compared with traditional supervised learning, ZSL is more challenging because the model must rely on shared semantic information such as attribute vectors or natural language descriptions to transfer knowledge from seen to unseen classes (Lampert et al., 2009; Socher et al., 2013; Xian et al., 2017).

Existing approaches can be broadly divided into two classes. Generative methods synthesize features of unseen classes via conditional generative models (Xian et al., 2018; Felix et al., 2018; Schonfeld et al., 2019), which alleviate the rigidity of prototypes but often suffer from training instability, mode collapse, and high computational cost. In contrast, non-generative methods typically construct class-level semantic prototypes and directly align visual features to these fixed prototypes (Akata et al., 2015; Zhang & Saligrama, 2016; Xie et al., 2019). Although simple and efficient, such methods assume that a single static prototype per class is sufficient, overlooking the considerable intra-class variation among instances. For example, different birds of the same species may vary in plumage color, pose, sex, or background context. A fixed class prototype cannot capture these differences simultaneously, leading to semantic drift, i.e., a mismatch between the instance semantics and the class prototype. As illustrated in Figure 1, individuals within the same class

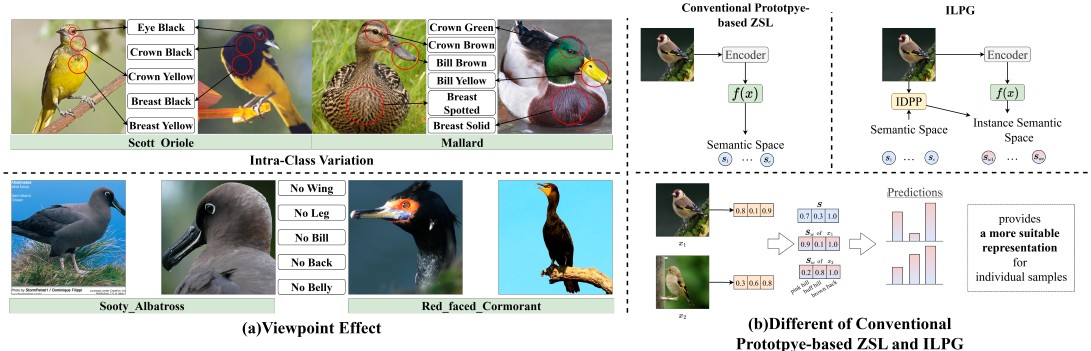

Figure 1: Individuals within the same class may exhibit diverse attributes due to factors such as gender, age, or viewpoint (e.g., "Crown Yellow" vs. "Crown Black," or when only head attributes are visible). If features are forcibly aligned to a single class-level semantic vector, the attention mechanism tends to emphasize only shared attributes while neglecting individual variations.

can exhibit diverse attributes (e.g., differences in color, pose, sex, or viewpoint). If all samples are forced to align to a single shared class-level prototype, the model tends to emphasize only the common attributes while neglecting instance- specific variations. This mismatch between sample-level semantics and class-level prototypes naturally leads to semantic drift and degraded recognition accuracy.

To address this limitation, some studies have introduced subspace methods or multi-prototype methods (Yu et al., 2020; Wang & et al., 2021; Liu & et al., 2021), enabling each class to be represented by multiple vectors instead of a single one. While this increases representational flexibility, the prototypes are still shared across all instances within a class rather than personalized for each image. In other words, the model continues to assume that all samples of the same class can be uniformly described in the semantic space, which makes it difficult to fully capture instance-level heterogeneity. This inherent limitation contributes to a performance bottleneck: when the attributes expressed by a particular instance deviate from the shared class prototype, recognition accuracy suffers. Motivated by these observations, we propose the *Instance-Level Prototype Generation* (ILPG) framework, which explicitly generates a personalized semantic prototype for *each input instance*. ILPG consists of three key components: (1) *Attribute-to-Visual Localization* (AVL) module, which employs cross-attention to localize attribute-relevant visual tokens, (2) the *Instance-Driven Prototype Personalization* (IDPP) module, which adjusts the static class prototype through instance-specific weights, and (3) the *Attribute Prototype Classifier* (APC), which performs semantic matching and classification based on the personalized prototypes. This design preserves the global semantic meaning of each class while introducing instance-level flexibility, thereby mitigating semantic drift.

Our main contributions are summarized as follows:

- We propose ILPG, a lightweight framework for instance-level prototype personalization, which dynamically generates a semantic prototype for each sample to better align visual and semantic spaces.
- We design a suite of complementary loss functions *Instance Prototype Matching* (IPM), *Semantic-Aware Self-Calibration* (SAS), *Visual–Semantic Residual Alignment* (VSR), and *Prototype Contrast Refinement* (PCR) to jointly encourage discriminability and semantic consistency.
- We conduct extensive experiments and ablation studies on three standard benchmarks (CUB, SUN, and AWA2) and further evaluate ILPG with different frozen encoders (e.g., DINOv2 (Caron et al., 2021), CLIP (Radford et al., 2021), SimCLR (Chen et al., 2020)). Results show that ILPG consistently improves zero-shot recognition performance under diverse settings and provides interpretable attribute alignment.

## 2 RELATED WORK

**Generative approaches.** A major line of work in ZSL employs generative models to synthesize features of unseen classes. Early efforts leverage GANs or VAEs to generate visual embeddings conditioned on semantic vectors, effectively reducing ZSL to a conventional supervised classification problem (Xian et al., 2018; Felix et al., 2018; Schonfeld et al., 2019). These methods alleviate the prototype rigidity by augmenting unseen data, but they often suffer from high computational cost, unstable training, and mode collapse. More recent studies explore diffusion models or large-scale generative priors (Ye et al., 2024), which further improve synthesis quality. Nevertheless, the reliance on synthetic features introduces additional complexity and may still fail to capture fine-grained instance-specific variations.

**Non-generative prototype-based approaches.** In contrast, non-generative methods directly construct semantic prototypes and align them with visual features. Classic works embed instances into a joint semantic space and assign labels via nearest-neighbor matching (Akata et al., 2015; Zhang & Saligrama, 2016). While computationally efficient, such approaches typically adopt a single static prototype per class, thereby ignoring intra-class variability such as pose, background, or viewpoint changes. Subsequent improvements incorporate attention mechanisms to enhance local region–attribute correspondence (Xie et al., 2019; Chen et al., 2022a), which improve fine-grained alignment but still rely on shared class-level prototypes.

**Multi-prototype and subspace approaches.** To address the limitations of static prototypes, several studies propose to represent each class with multiple prototypes or learn a subspace spanned by basis vectors (Yu et al., 2020; Wang & et al., 2021; Liu & et al., 2021; Song & et al., 2024; Jiang & et al., 2024). These designs increase flexibility by capturing diverse visual patterns within a class, yet the resulting prototypes remain *class-shared*. In practice, when an individual sample deviates from these shared prototypes, semantic drift still occurs, which limits the effectiveness of recognition in challenging scenarios.

**Position of our work.** Our ILPG framework departs from the prevailing "one (or a few) prototypes per class" paradigm. Instead, it generates a unique, instance-adaptive prototype for every input image. Through the *Attribute-to-Visual Localization* (AVL) module and instance-driven personalization, ILPG explicitly models intra-class heterogeneity while retaining the global semantic consistency of classes. This design fundamentally distinguishes ILPG from prior multi-prototype or subspace methods, providing a principled solution to prototype rigidity and semantic drift.

## 3 PROPOSED METHOD

**Notation and Problem Formulation** We begin by introducing the notation and problem formulation. Let the training set be $\boldsymbol{D}_s = \{(x_i^s, y_i^s)\}_{i=1}^{N_s}$ consisting of $\boldsymbol{C}_s$ seen classes, where $x_i^s \in \boldsymbol{X}$ denotes the $i$-th image and $y_i^s \in \boldsymbol{Y}_s$ its corresponding class label. An additional collection of unseen classes $\boldsymbol{C}_u$ is represented by the unlabeled set $\boldsymbol{D}_u = \{(x_i^u, y_i^u)\}_{i=1}^{N_u}$, with $x_i^u \in \boldsymbol{X}$ the unseen-class images and $y_i^u \in \boldsymbol{Y}_u$ their labels. For every class $c \in \boldsymbol{C}_s \cup \boldsymbol{C}_u = \boldsymbol{C}$ we are given a class-level semantic vector $\boldsymbol{S} = [s_1, \ldots, s_N]^\top = \varphi(y), \boldsymbol{S} \in \mathbb{R}^{N \times d_s}$ containing $N$ categories that serve to transfer knowledge from seen to unseen classes. To adapt these semantics to the requirements of individual instances, we further learn an instance-aware semantic-adjustment weight vector $\boldsymbol{W} = [w_1, \ldots, w_N]^\top, \boldsymbol{W} \in \mathbb{R}^{N \times d_s}$ that refines $\boldsymbol{S}$ for each specific classification task. Moreover, we leverage language-model–based semantic attribute word embeddings $\boldsymbol{A} = [\boldsymbol{a}_1, \boldsymbol{a}_2, \ldots, \boldsymbol{a}_{d_s}]^\top, \boldsymbol{A} \in \mathbb{R}^{d_s \times d_a}$ pre-trained on attribute names Pennington et al. (2014). The goal of ZSL is to train on the seen classes $\boldsymbol{D}_s$ and test on the unseen classes $\boldsymbol{D}_u$. In the Conventional ZSL (CZSL) setting we predict $y^u \in \boldsymbol{Y}_u$, whereas in the Generalized ZSL (GZSL) setting we predict $y \in \boldsymbol{Y} = \boldsymbol{Y}_s \cup \boldsymbol{Y}_u$, with $\boldsymbol{Y}_s \cap \boldsymbol{Y}_u = \varnothing$.

**Conventional Prototype-Based ZSL** In conventional ZSL, classification is modeled as similarity matching between image embeddings and class-level semantic prototypes (see Algorithm 1). Given an image $x$, the

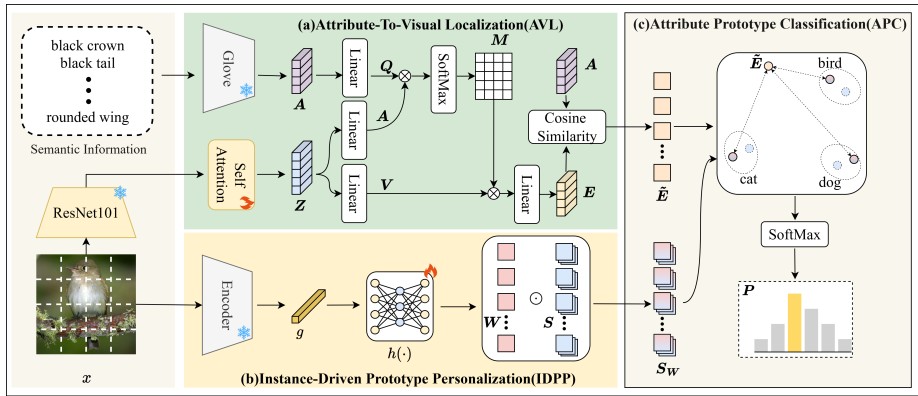

Figure 2: Overall architecture of the proposed ILPG model, consisting of *Attribute-to-Visual Localization* (AVL), *Instance-Driven Prototype Personalization* (IDPP), and *Attribute Prototype Classification* (APC).

encoder extracts $e = f(x)$, and each class $c$ is represented by a fixed prototype $S(c)$ from attributes or text. The prediction is:

$$\boldsymbol{P}(y = c|x) = \frac{\exp\langle e, S(c)\rangle}{\sum_{j \in C}\exp\langle e, S(j)\rangle}. \tag{1}$$

This paradigm enables knowledge transfer from seen to unseen classes but has limitations: (1) one static prototype per class ignores intra-class diversity, (2) prototypes lack adaptivity to individual samples, and (3) the gap between visual and semantic spaces often causes semantic drift. To overcome these issues, we propose ILPG (see Algorithms 2 and 3), which constructs dynamic, instance-specific prototypes to preserve semantic alignment while capturing visual diversity, alleviating prototype rigidity and semantic drift.

**Attribute-to-Visual Localization** As shown in Figure 2(a), We use a frozen ResNet101 network as our backbone to extract image features $\boldsymbol{I} = [\boldsymbol{i}_1, \boldsymbol{i}_2, \ldots, \boldsymbol{i}_R]^\top \in \mathbb{R}^{R \times d_v}$. And we adopt the widely-validated encoder–decoder architecture as our backbone. The core function of this module is to use attribute features as guidance, driving the attention mechanism to focus on visual regions that correspond to the attributes, thereby achieving attribute localization. In short: We encode the attribute names with the pre-trained language model GloVe to obtain the attribute vectors $\boldsymbol{A}$.

A cross-attention mechanism is employed to localize, within the visual features, the regions that match the attribute-guided queries. Let $\boldsymbol{Q}$, $\boldsymbol{K}$, and $\boldsymbol{V}$ denote Query, Key, and Value, respectively, defined as

$$\boldsymbol{Q} = \boldsymbol{A}\boldsymbol{W}_q, \quad \boldsymbol{K} = \boldsymbol{I}\boldsymbol{W}_k, \quad \boldsymbol{V} = \boldsymbol{I}\boldsymbol{W}_v, \tag{2}$$

where $\boldsymbol{W}_q \in \mathbb{R}^{d_a \times d}, \boldsymbol{W}_k, \boldsymbol{W}_v \in \mathbb{R}^{d_v \times d}$ are learnable projection matrices. The relevance between the query $\boldsymbol{Q}$ and the key $\boldsymbol{K}$ is computed to yield an attention map $\boldsymbol{M}$, which is normalized by Softmax. Finally, the attribute-localized features $\boldsymbol{E}$ are obtained by weighting the values $\boldsymbol{V}$ with the attention map:

$$\boldsymbol{E} = \boldsymbol{M}\boldsymbol{V} = \text{Softmax}\left(\frac{\boldsymbol{Q}\boldsymbol{K}^\top}{\sqrt{d}}\right)\boldsymbol{V}, \boldsymbol{E} \in \mathbb{R}^{d_s \times d_a}. \tag{3}$$

We compute the cosine similarity between the attribute-localized features $\boldsymbol{E}$ and the original attribute vectors $\boldsymbol{A}$, thereby transforming E into a feature $\tilde{\boldsymbol{E}}$ with the same dimensionality as the class prototypes, facilitating classification. This can be understood as calculating a score for each attribute, where the class prototype is composed of the scores of all attributes within that class, reflecting the specific attributes and their specific

representations for that class.

$$\tilde{E} = \frac{E \cdot A^T}{\|E\|\|A\|}, \tilde{E} \in \mathbb{R}^{d_s}. \tag{4}$$

**Instance-Driven Prototype Personalization.** As shown in Figure 2(b), $\tilde{E}$ produced by the AVL module has already performed a preliminary localization of sample attributes with the aid of the attention mechanism. However, inherent inter-sample variations still introduce localization bias. To mitigate this effect, we propose the IDPP module, which injects extra instance-level cues and dynamically generates the most compatible class prototype for each individual image, thereby boosting classification performance. Owing to the intrinsic within-class collapsing tendency of classification models, features of the same class are pulled toward the class prototype, weakening inter-instance distinctiveness. We therefore introduce a frozen DINOv2 encoder to provide an instance-specific signature $g$ (the reason for choosing DINOv2 is given in the experiments). To exploit the instance-difference information embodied in $g$, we design the Instance Semantic Generator (ISG), a lightweight MLP that converts $g$ into personalized weights $W$. $W = h(g)$ to fuse g with the original class semantic vector S while preventing catastrophic forgetting of inter-class knowledge, we introduce Dual-Path Tuning (DPT), DPT balances a personalization path $W \odot S$ and a fidelity path $S$, yielding prototypes that are both instance-customized and cross-domain transferable.

$$S_W = W \odot S + S, S_W \in \mathbb{R}^{N \times d_s}. \tag{5}$$

**Attribute Prototype Classification** APC performs one-instance-one-prototype voting: for each image, its $\tilde{E}$ is matched against its own generated prototype set $\{S_W^{(i)}\}_{i=1}^C$ via row-wise dot product, e taking the dot product of each row of the matrix with the vector and summing all the.

$$P_i = \frac{\exp\langle \tilde{E}^\top, S_W^{(i)} \rangle}{\sum_{j=1}^C \exp\langle \tilde{E}^\top, S_W^{(j)} \rangle}, \tag{6}$$

where $\langle \cdot, \cdot \rangle$ denotes row-wise dot-sum. Thus, every test image casts a similarity vote only on its personally generated prototypes, yielding the final class distribution $P_i \in \Delta^C$.

**Model Optimization** To ensure instance-level prototype generation while maintaining cross-domain transferability, ILPG is trained with four synergistic losses, *Instance–Prototype Matching Loss* (IPM) *Semantic-Aware Self-Calibration Loss* (SAS), *Visual–Semantic, Residual Alignment Loss* (VSR), *Prototype Contrast Refinement Loss* (PCR) Together they guarantee that the model converges on seen classes and generalizes smoothly to unseen ones.

**Instance Prototype Matching Loss** Although prototype classification is common in ZSL, prior work relies on static prototypes. IDPP instead generates a unique prototype per image, so an instance-level matching loss is required. Given image $x$ and its dynamic prototype set $\{S_W^{(j)}\}_{j=1}^C$, IPM maximizes the posterior probability of the ground-truth prototype via standard cross-entropy:

$$\mathcal{L}_{\text{IPM}} = -\sum_{(x,y)\in D} \log \frac{\exp\langle \tilde{E}(x)^\top S_W^{(y)} \rangle}{\sum_{i\in C} \exp\langle \tilde{E}(x)^\top S_W^{(i)} \rangle}, \tag{7}$$

$y$ denotes the ground-truth class label of the current sample the correct class assigned to the image in the training set.

**Semantic Aware Self-Calibration Loss** Vanilla self-calibration applies a uniform pulling force to all unseen classes, causing the semantic space to be over-smoothed and damaging the seen-unseen manifold. We

propose SAS, which replaces this crude strategy with a sample-adaptive gating mechanism. For each seen-class sample $(x_i, y_i)$ and every unseen class $c \in \boldsymbol{C}_u$, we compute a semantic reachability weight

$$w_{y_i,c} = \frac{1 + \cos(\boldsymbol{S}_{y_i}, \boldsymbol{S}_c)}{2} \in [0, 1], \tag{8}$$

that varies per sample-class pair. The resulting objective is

$$\mathcal{L}_{\text{SAS}} = -\sum_{i=1}^{n} \sum_{c \in \boldsymbol{C}_u} w_{y_i,c} \, \log \frac{\exp\langle \tilde{\boldsymbol{E}}^{\top} \boldsymbol{S}_W^{(c)} \rangle}{\sum_{j=1}^{\boldsymbol{C}} \exp\langle \tilde{\boldsymbol{E}}^{\top} \boldsymbol{S}_W^{(j)} \rangle}, \tag{9}$$

SAS boosts probable unseen classes while suppressing semantically distant noise

**Visual Semantic Residual Alignment Loss** IDPP adjusts prototypes via Hadamard residual, yet the residual may drift freely. VSR explicitly constrains the $L_2$ distance between the fused feature and its corresponding prototype, ensuring that instance-level changes stay on the transferable manifold:

$$\mathcal{L}_{\text{VSR}} = \sum_{i \in \boldsymbol{C}_s} \left\| \tilde{\boldsymbol{E}}(\boldsymbol{x}_i) - \boldsymbol{S}_W^{(y_i)} \right\|_2^2, \tag{10}$$

VSR acts as a steady-state regularizer that preserves cross-domain semantic integrity.

**Prototype Contrast Refinement Loss** After adjustment, $\boldsymbol{S}_W$ must still faithfully anchor to the original semantics. PCR treats the static prototype $\boldsymbol{S}^{(y)}$ as the positive anchor and all other class prototypes as negatives, pulling $\boldsymbol{S}_W^{(y)}$ closer while pushing it away from others, where $\tau$ is a temperature hyper-parameter.

$$\mathcal{L}_{\text{PCR}} = -\log \frac{\exp\big(\langle \boldsymbol{S}_W^{(y)}, \boldsymbol{S}^{(y)} \rangle / \tau\big)}{\sum_{i \in \boldsymbol{C}} \exp\big(\langle \boldsymbol{S}_W^{(y)}, \boldsymbol{S}^{(i)} \rangle / \tau\big)}. \tag{11}$$

Finally, we define the overall loss function of ILPG as,

$$\mathcal{L}_{\text{total}} = \mathcal{L}_{\text{IPM}} + \lambda_{\text{SAS}} \mathcal{L}_{\text{SAS}} + \lambda_{\text{VSR}} \mathcal{L}_{\text{VSR}} + \lambda_{\text{PCR}} \mathcal{L}_{\text{PCR}}, \tag{12}$$

where $\lambda_{\text{SAS}}$, $\lambda_{\text{VSR}}$, and $\lambda_{\text{PCR}}$ are weighting coefficients that balance the contributions of the corresponding loss terms. We theoretically and empirically verify the robustness of ILPG under prototype drift and semantic shift, detailed guarantees and proofs are deferred to the Appendix A.

## 4 THEORETICAL ANALYSIS

Beyond empirical performance, ILPG also enjoys theoretical guarantees. We analyze its behavior under prototype drift and semantic shift, and compare it with a baseline method. Full proofs are deferred to Appendix B.

**Theorem 1** (Robustness Radius Dominance). Under standard Lipschitz and margin assumptions, ILPG achieves a robustness radius $r_{\text{safe}}^{\text{ILPG}} \geq r_{\text{safe}}^{\text{Base}}$, implying that ILPG is provably no less robust to prototype drift than the baseline.

**Sketch.** Let $\Delta(y)$ denote the drift of class-$y$ prototypes. For any competitor $j \neq y$, the prediction gap satisfies

$$g_y(\tilde{E}, S_W) - g_j(\tilde{E}, S_W) \geq \gamma(x) - L(\Delta(y) + \Delta(j)),$$

Table 1: Comparison results on CUB, SUN and AWA2 (Backbone column added in second column).

| Methods | Backbone | CUB | | | | SUN | | | | AWA2 | | | |
|---------|----------|------|------|------|------|------|------|------|------|------|------|------|------|
| | | CZSL | U | S | H | CZSL | U | S | H | CZSL | U | S | H |
| GEM-ZSL Liu et al. (2021) | ResNet101 | 77.8 | 64.8 | 77.1 | 70.4 | 62.8 | 38.1 | 35.7 | 36.9 | 67.3 | 64.8 | 77.5 | 70.6 |
| SE-ZSL Kim et al. (2022) | ResNet101 | – | 53.1 | 60.3 | 56.4 | – | 45.8 | 40.7 | 43.1 | – | 59.9 | 80.7 | 68.8 |
| MSDN Chen et al. (2022b) | ResNet101 | 76.1 | 68.7 | 67.5 | 68.1 | 65.8 | 52.2 | 34.2 | 41.3 | 70.1 | 62.0 | 74.5 | 67.7 |
| TransZero Chen et al. (2022a) | ResNet101 | 76.8 | 69.3 | 68.3 | 68.8 | 65.6 | 52.6 | 33.4 | 40.8 | 70.1 | 61.3 | 82.3 | 70.2 |
| ICIS Christensen et al. (2023) | ResNet101 | 60.6 | 45.8 | 73.7 | 56.5 | 51.8 | 45.2 | 25.6 | 32.7 | 64.6 | 35.6 | 93.3 | 51.6 |
| DUET Chen et al. (2023) | ViT-Base | 72.3 | 62.9 | 72.8 | 67.5 | 64.4 | 45.7 | 45.8 | 45.8 | 69.9 | 63.7 | 84.7 | 72.7 |
| PSVMA Liu et al. (2023) | ViT-Base | – | 70.1 | 77.8 | 73.8 | – | 61.7 | 45.3 | 52.3 | – | 73.6 | 77.3 | 75.4 |
| ReZSL Ye et al. (2023) | ResNet101 | 80.9 | 72.8 | 74.8 | 73.8 | 63.0 | 47.4 | 34.8 | 40.1 | 70.9 | 63.8 | 85.6 | 73.1 |
| DML Zhang et al. (2024) | ViT-Base | – | 57.1 | 81.6 | 67.2 | – | 39.6 | 52.7 | 45.9 | – | 62.2 | 82.3 | 70.9 |
| ZSLViT Chen et al. (2024) | ViT-Base | 78.9 | 69.4 | 78.2 | 73.6 | 68.3 | 45.9 | 48.4 | 47.3 | 70.7 | 66.1 | 84.7 | 74.2 |
| ILPG (Ours) | ResNet101 | 81.1 | 71.3 | 80.5 | 75.6 | 72.5 | 61.7 | 49.7 | 55.1 | 71.7 | 64.3 | 86.1 | 73.6 |

where $\gamma(x)$ is the margin at zero drift and $L$ is the Lipschitz constant. Thus, correctness is preserved if $2Lr < \gamma(x)$ with drift radius $r$. Since ILPG guarantees smaller or equal drift compared to the baseline, its robustness radius is no smaller.

**Theorem 2** (Unseen Risk Dominance). Given comparable hypothesis complexity, ILPG achieves a no-larger unseen risk bound under semantic shift than the baseline.

**Sketch.** The calibration mechanism of ILPG ensures tighter alignment between adapted prototypes and semantic anchors, leading to lower weighted seen-class loss. By transferring this advantage via the calibration coupling, the unseen-class risk bound for ILPG is provably no larger than that of the baseline.

**Consequence.** Combining Theorems 6 and 7, we conclude that ILPG is theoretically guaranteed to be at least as robust and generalizable as the baseline under prototype drift and semantic shift. Complete derivations and proofs are provided in Appendix B.

## 5 EXPERIMENTS

**Experimental Setup** We evaluate ILPG on three ZSL benchmarks: CUB Welinder et al. (2010) (fine-grained birds), SUN Patterson & Hays (2012) (fine-grained scenes), and AWA2 Xian et al. (2017) (coarse-grained animals), covering both fine- and coarse-grained scenarios. Following the protocol of Xian et al. Xian et al. (2017), we report average per-class Top-1 accuracy. In CZSL, only unseen-class images are evaluated. In GZSL, both seen and unseen classes are tested, and we compute seen accuracy (S), unseen accuracy (U), and their harmonic mean $H = 2US/(U + S)$. We implement ILPG in PyTorch using DINOv2 Oquab et al. (2023) as the visual encoder and GloVe Pennington et al. (2014) as the semantic embedding. Training is performed for 100 epochs with SGD (momentum 0.9, weight decay $1 \times 10^{-4}$), initial learning rate $1 \times 10^{-4}$, and batch size 50. All experiments run on a single NVIDIA RTX 4090 GPU.

Table 2: Ablation results (%) on CUB and SUN.

| Methods | CUB | | SUN | |
|---------|------|------|------|------|
| | acc | H | acc | H |
| Baseline | 61.9 | 33.9 | 61.1 | 34.2 |
| ILPG w/o SAS | 80.8 | 48.0 | 65.8 | 44.3 |
| ILPG w/o VSR | 79.1 | 59.3 | 66.3 | 43.3 |
| ILPG w/o PCR | 77.8 | 68.9 | 66.8 | 44.2 |
| ILPG w/o IDPP | 73.1 | 62.7 | 66.7 | 44.6 |
| ILPG | 81.1 | 75.6 | 72.5 | 55.1 |

**Comparison with State-of-the-Art Methods** We compare ILPG with state-of-the-art methods under both CZSL and GZSL settings (Table 1). On fine-grained datasets such as **CUB** and **SUN**, ILPG achieved the best Top-1 accuracy (81.1%) and harmonic mean (75.0%) on CUB, and Top-1 accuracy (72.5%) and harmonic mean (55.1%) on SUN, showing clear advantages. This improvement is mainly attributed to the AVL module, which highlights attribute-relevant regions, and the IDPP module, which personalizes prototypes to reduce intra-class confusion (such as pose, viewpoint, and background), demonstrating robustness in complex scene understanding. The SAS

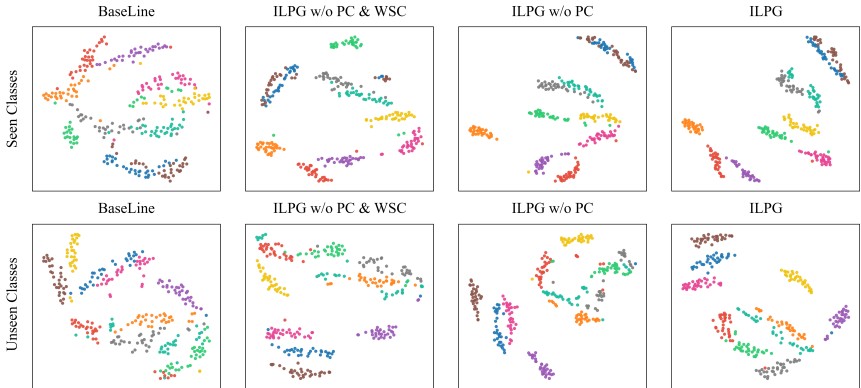

Figure 3: t-SNE visualization of class-level features: Feature distributions for seen classes and unseen classes, respectively. The features are extracted from: (1) Baseline, (2) ILPG w/o PCR & SAS, (3) ILPG w/o PCR, and (4) ILPG. The 10 colors correspond to 10 randomly selected seen/unseen classes from the CUB dataset.

and PCR modules effectively mitigate semantic drift in long-tail classes. On the coarse-grained **AWA2** dataset, ILPG remains competitive (71.7%, H=73.6%), although improvements are smaller since there is less intra-class variation in coarse-grained categories, and ILPG naturally faces a disadvantage on coarse-grained datasets. Efficiency analysis further shows that, as presented in Table 3, we conducted efficiency experiments on the modules involved in training. Removing IDPP reduces the number of parameters (from 3.67M to 3.31M), latency (from 7.37ms to 7.07ms), while causing only a 4.2% throughput gap, and keeping memory usage unchanged (0.90GB). This confirms that IDPP brings significant accuracy improvements with negligible overhead. To further demonstrate our efficiency, we selected two comparative models and presented their running efficiency compared to our model in Table 4. Compared to ZSLViT and ReZSL, ILPG has significantly fewer parameters. Specifically, ILPG only adds an MLP within a commonly used zero-shot framework to generate instance-level class prototypes, allowing us to achieve competitive results with fewer parameters. ILPG also shows a clear advantage in FLOPs, mainly due to the operation of generating class prototypes. Although the Lat. (latency) of ILPG is slightly higher than that of ReZSL, in terms of memory usage (Mem.) and throughput (Thr.), ILPG is far ahead. Especially in throughput, ILPG can handle more image samples, demonstrating higher inference efficiency.

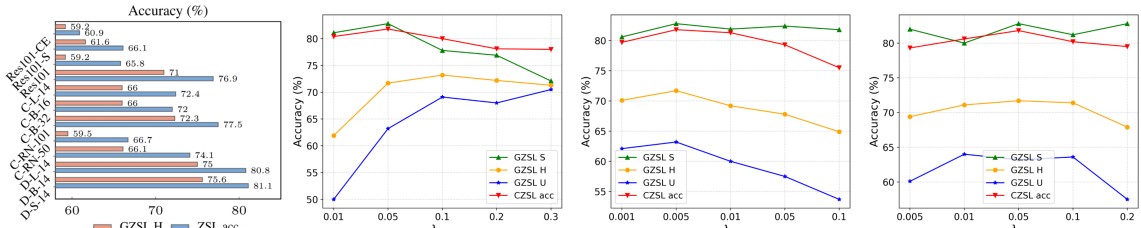

Figure 4: Ablation study of different encoders on CUB.

Figure 5: Hyperparameter analysis of $\lambda_{SAS}$, $\lambda_{VSR}$ and $\lambda_{PCR}$. We show the CZSL and GZSL performance variations on CUB.

**Ablation Study** We conduct ablation experiments on CUB and SUN to assess the contribution of each component. As shown in Table 2, the baseline encoder–decoder with instance–prototype matching yields poor performance (H=33.9 on CUB). Removing IDPP, PCR, VSR, or SAS in turn consistently degrades results, with IDPP having the largest impact (H=62.7). This confirms the importance of instance-level prototype personalization. Similar trends are observed on SUN, where the full ILPG improves H from 34.2 (baseline) to 55.1. We further evaluate different frozen encoders within IDPP, including DINOv2, CLIP, and

ResNet101 variants (Figure 4). Interestingly, DINOv2-ViT-S/14 achieves the best overall results (ZSL=81.1, H=75.6), outperforming larger DINOv2 (B/14, L/14) and CLIP models, suggesting that moderate-capacity self-supervised ViTs better preserve instance-level diversity crucial for ILPG. For ResNet101, we compare three variants: the original backbone, supervised fine-tuning with cross-entropy, and unsupervised fine-tuning with SimCLR. Results show that SimCLR (H=66.1) clearly outperforms both the vanilla backbone (H=59.2) and the cross-entropy fine-tuned version (H=59.2), indicating that unsupervised contrastive learning preserves richer instance-level cues, whereas supervised training tends to collapse intra-class variations.

Overall, these results highlight that ILPG benefits from both its synergistic loss design and the use of self-supervised visual encoders. To further analyze classification behavior, we provide confusion matrices in Appendix E.To further evaluate the robustness of ILPG, we conduct systematic occlusion experiments on CUB with varying occlusion ratios. As detailed in Appendix F, the results consistently show that ILPG outperforms ILPG-IDPP, especially under medium to heavy occlusion, demonstrating that the instance-level dynamic prototypes of ILPG effectively mitigate information loss and preserve recognition performance.

Table 3: Inference efficiency FP16 b=32 $224^2$.

| Metric | ILPG–IDPP | ILPG (Full) |
|---|---|---|
| Params (M) | 3.31 | 3.67 |
| Lat. (ms) | $7.07 \pm 0.30$ | $7.37 \pm 0.22$ |
| Thr. (img/s) | 4527.78 | 4339.37 |
| Mem. (GB) | 0.90 | 0.90 |

Table 4: Performance Metric Comparison Experiment

| Method | Params (M) | FLOPs (MACs) (M) | Lat. (ms) | Mem. (GB) | Thr. (img/s) @ batch=50 |
|---|---|---|---|---|---|
| ZSLViT | 95.871 | 50652.970 | 17.411 | 0.441 | 53.394 |
| ReZSL | 88.709 | 16851.515 | 6.366 | 0.352 | 150.458 |
| ILPG | 68.224 | 13769.496 | 15.637 | 0.283 | 245.52 |

**t-SNE Visualizations.** As shown in Figure 3, we present the t-SNE visualization (Maaten and Hinton 2008) of classification features extracted on the CUB dataset for both seen and unseen classes. As the IDPP, PCR, and SAS modules are progressively incorporated, the classification features become increasingly compact and inter-class overlap is markedly reduced. These results clearly demonstrate that the ILPG model effectively alleviates the misalignment between instance-level features and static class semantic vectors.

We also visualized our category prototypes. Using the CUB dataset, we performed t-SNE visualization of the category prototypes during the training process. As shown in Figure 7, we present images at three stages: the beginning of training, the middle of training, and the end of training. It can be observed that, due to our thoughtful design, the category prototypes consistently maintain the correct semantic structure throughout the training process. Additionally, as training progresses, the category prototypes shift within a certain range, which aligns with our goal of better aligning instance-level category prototypes with image features.

**Visualization of Attention Maps.** To intuitively demonstrate the substantial advantage of instance-level dynamic semantic vectors in aligning with instance visual features, we visualize the attention maps of the baseline (without instance-level dynamic semantic vectors) and our ILPG. To better highlight their differences, we group attributes by body part and jointly normalize the attention weights within each part, thereby emphasizing which attributes the model actually focuses on within that part.

As shown in Figure 6, The baseline attention of model is limited to a coarse localization of body parts and fails to distinguish among different attributes within the same part. In contrast, ILPG leverages instance-level dynamic semantic vectors to enable fine-grained attribute discrimination, precisely pinpointing the specific attribute values that truly exist in a body part while effectively suppressing interference from absent attributes. Supplementary attention-weight visualizations for unseen classes on CUB are provided in Appendix C.

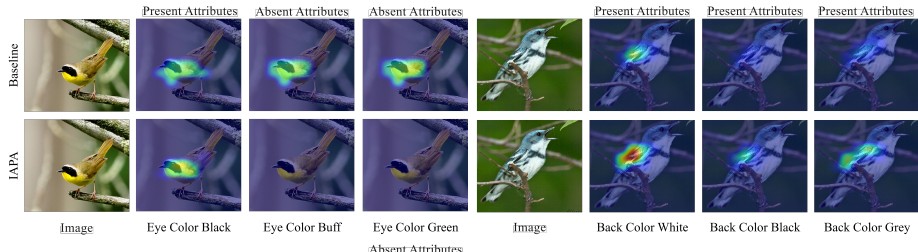

Figure 6: Visualization of attention maps for the baseline and our ILPG.

**Hyperparameter Analysis** Our method involves three key hyper-parameters: $\lambda_{\text{SAS}}$, $\lambda_{\text{VSR}}$, and $\lambda_{\text{PCR}}$. Extensive experiments on CUB (as shown in Fig. 5) show that even when these weights vary across orders of magnitude, the performance remains stable, indicating that ILPG is insensitive to hyper-parameter settings. This robustness arises from the complementary roles of the three losses, SAS improves the separability of seen and unseen classes, VSR enforces cross-domain consistency, and PCR strengthens the link between personalized prototypes and global semantics, mitigating semantic drift. For CUB, we set $\lambda_{\text{SAS}} = 0.05$, $\lambda_{\text{VSR}} = 0.005$, and $\lambda_{\text{PCR}} = 0.05$. Further analyses for SUN and AWA2 are provided in Appendix G.

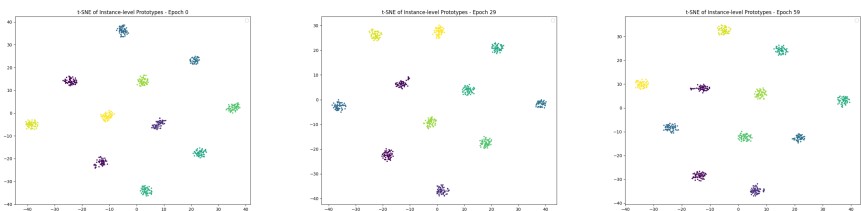

Figure 7: Category Prototype t-SNE Visualization

## 6 CONCLUSION

In this paper, we propose Instance-Level Prototype Generation (ILPG) for ZSL. ILPG is based on two main components: AVL and APC. Their primary function is to fuse attribute features with the visual features of samples to obtain integrated features. To facilitate better alignment between the integrated features and class semantic vectors, ILPG introduces the IDPP module. The IDPP converts features extracted by a pre-trained encoder, which contain instance-specific differences, into instance-level adjustment weights. These weights are then used to adjust the class semantic vectors at the instance level, enabling precise alignment. Moreover, ILPG proposes two loss functions: SAS and PCR. The SAS fully utilizes the class similarity information preserved in the class semantic vectors, while the PCR promotes accurate learning of instance-level class prototypes. Experiments on three key datasets confirm the effectiveness of our method.

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

# A  APPENDIX

# A  FUNDAMENTAL DERIVATIONS

**Notation.** $Y_s, Y_u$ denote seen and unseen class sets. For class $c$, $S(c) \in \mathbb{R}^d$ with $\|S(c)\|_2 = 1$. Weights $w_{y,u} = \langle S(y), S(u) \rangle \in [0,1]$, $\sum_{y \in Y_s} w_{y,u} = 1$. $D_y$ (resp. $D_u$) is the class-conditional distribution of seen (resp. unseen) class. Hypotheses $f \in F$ produce logits, per-class loss $L_c(x) = -\log p_c(x) \in [0, M]$. $\mathrm{TV}(P, Q) = \sup_A |P(A) - Q(A)|$. $\widehat{\mathbb{E}}_{D_y}$ is the empirical mean over $n_y$ i.i.d. samples.

## A.1  BACKGROUND ON RADEMACHER COMPLEXITY

**Definition 1** (Rademacher complexity). Let $\sigma = (\sigma_1, \ldots, \sigma_n)$ be i.i.d. Rademacher variables (uniform on $\{\pm 1\}$). For sample $S = (x_1, \ldots, x_n)$ and function class $F$,

$$\widehat{\mathcal{R}}_S(F) = \mathbb{E}_\sigma \left[ \sup_{f \in F} \frac{1}{n} \sum_{i=1}^n \sigma_i f(x_i) \right], \quad \mathcal{R}_n(F) = \mathbb{E}_S[\widehat{\mathcal{R}}_S(F)].$$

**Lemma 1** (Symmetrization). For bounded loss $\ell \in [0, M]$ and $f \in F$,

$$\mathbb{E}_D \ell(f) - \widehat{\mathbb{E}}_D \ell(f) \leq 2\,\mathcal{R}_n(\ell \circ F),$$

where $\ell \circ F = \{x \mapsto \ell(f(x)) : f \in F\}$.

**Lemma 2** (Contraction). If $\phi : \mathbb{R} \to \mathbb{R}$ is $L$-Lipschitz with $\phi(0) = 0$, then

$$\widehat{\mathcal{R}}_S(\phi \circ F) \leq L\,\widehat{\mathcal{R}}_S(F).$$

These imply standard high-probability generalization bounds: for any $\delta \in (0, 1)$, with probability at least $1 - \delta$,

$$\forall f \in F : \ \mathbb{E}_D \ell(f) \leq \widehat{\mathbb{E}}_D \ell(f) + 2\,\mathcal{R}_n(F) + M\sqrt{\frac{\log(2/\delta)}{2n}}.$$

## A.2  THEOREM 1: GENERALIZATION BOUND OF SAS

**Lemma 3** (Calibration coupling). Assume there exists $L > 0$ such that for any $u$,

$$\Big| \mathbb{E}_{D_u} L_u - \sum_{y \in Y_s} w_{y,u}\, \mathbb{E}_{D_y} L_y \Big| \leq L \sum_{y \in Y_s} w_{y,u}\, \mathrm{TV}(D_u, D_y).$$

**Theorem 3** (SAS generalization bound). For any unseen class $u$ and $\delta \in (0, 1)$, with probability $\geq 1 - \delta$,

$$\mathbb{E}_{D_u} L_u \leq \sum_{y \in Y_s} w_{y,u} \left( \widehat{\mathbb{E}}_{D_y} L_y + 2\,\mathcal{R}_{n_y}(F) + M\sqrt{\frac{\log(2|Y_s|/\delta)}{2n_y}} \right)$$

$$+ L \sum_{y \in Y_s} w_{y,u}\, \mathrm{TV}(D_u, D_y). \tag{13}$$

If additionally $\mathrm{TV}(D_u, D_y) \leq c\|S(u) - S(y)\|_2$, then the last term can be replaced by $Lc \sum_{y \in Y_s} w_{y,u} \|S(u) - S(y)\|_2$.

*Proof.* Apply the Rademacher bound to each $D_y$ (symmetrization + contraction). Take the convex combination with weights $w_{y,u}$, then Lemma 3 completes the inequality. Union bound over $|Y_s|$ classes yields the stated confidence. $\qquad\square$

## A.3 THEOREM 2: SEMANTIC STABILITY OF PCR

$$L_{\mathrm{PCR}} = -\log \frac{\exp(\rho_y)}{\sum_{j \in \mathcal{C}} \exp(\rho_j)}, \quad \rho_j = S_W(y)^\top S(j).$$

**Assumption 1** (Bounded negatives). *There exists $\alpha < 1$ such that $\max_{j \neq y} \rho_j \leq \alpha$.*

**Lemma 4** (Lower bound on positive similarity). *If $L_{\mathrm{PCR}} \leq \varepsilon$, then*

$$\rho_y \geq \alpha + \log(C-1) - \log(e^\varepsilon - 1).$$

**Theorem 4** (Semantic stability). *If $L_{\mathrm{PCR}} \leq \varepsilon$ and $\varepsilon \leq \log\big(1 + (C-1)e^{\alpha-(1-\tau)}\big)$, then*

$$\|S_W(y) - S(y)\|_2 \leq \sqrt{2\tau}, \quad |S_W(y)^\top S(j) - S(y)^\top S(j)| \leq \sqrt{2\tau}, \ \forall j.$$

*Proof.* The loss bound implies $\rho_y$ exceeds a threshold. Imposing $\rho_y \geq 1 - \tau$ ensures cosine closeness. The $\ell_2$ bound follows from $\|a - b\|_2^2 = 2(1 - a^\top b)$ for unit vectors. □

## A.4 THEOREM 3: GEOMETRIC CONSISTENCY OF VSR

$$L_{\mathrm{VSR}} = \sum_{i \in C_s} \|E(x_i) - S_W(y_i)\|_2^2.$$

**Theorem 5** (Unique minimizer). *$L_{\mathrm{VSR}}$ is strictly convex in each $E(x_i)$, the unique minimizer is $E(x_i) = S_W(y_i)$ for all $i \in C_s$.*

*Proof.* Each term has Hessian $2I_d \succ 0$, hence strict convexity. Setting gradient $2(E(x_i) - S_W(y_i)) = 0$ yields the unique solution. □

**Remark 1.** This guarantees feature-prototype alignment in training. Residual parametrization $S_W(y) = S(y) \odot W + S(y)$ restricts adapted prototypes to stay near semantic anchors.

## B RELATIVE SUPERIORITY UNDER PROTOTYPE DRIFT / SEMANTIC SHIFT

**Setup.** Let $S(\cdot)$ be unit-norm class prototypes, and $S_W^{\mathrm{ILPG}}(y)$, $S_W^{\mathrm{Base}}(y)$ denote adapted prototypes produced by ILPG and a baseline, respectively, for class $y$. For an input instance $x$ with fused feature $\tilde{E}(x)$, define the (per-class) score/logit map $g_j(\tilde{E}, S_W) = \phi(\tilde{E}, S_W(j))$. We make two standard, verifiable assumptions:

**Assumption 2** (Lipschitz scores in prototype). *There exists $L > 0$ such that for all $j$, $\big|g_j(\tilde{E}, S_W) - g_j(\tilde{E}, S'_W)\big| \leq L \|S_W(j) - S'_W(j)\|_2$.*

**Assumption 3** (Zero-drift classification margin). *At zero drift (i.e., using the semantic anchor $S(\cdot)$), the ILPG classifier enjoys margin $\gamma_{\mathrm{ILPG}}(x) = g_y(\tilde{E}, S) - \max_{j \neq y} g_j(\tilde{E}, S) > 0$ for the ground-truth $y$, and the baseline has margin $\gamma_{\mathrm{Base}}(x)$ (not necessarily identical).*

**PCR/VSR-induced drift bounds.** Let $\Delta_{\mathrm{ILPG}}(y) = \|S_W^{\mathrm{ILPG}}(y) - S(y)\|_2$ and $\Delta_{\mathrm{Base}}(y) = \|S_W^{\mathrm{Base}}(y) - S(y)\|_2$. By Theorem A.3 (PCR stability) with the same negative-similarity budget $\alpha$ and Theorem A.4 (VSR alignment), if $L_{\mathrm{PCR}}^{\mathrm{ILPG}} \leq L_{\mathrm{PCR}}^{\mathrm{Base}}$ and $\mathcal{L}_{\mathrm{VSR}}^{\mathrm{ILPG}} \leq \mathcal{L}_{\mathrm{VSR}}^{\mathrm{Base}}$, then for any $\tau \in (0, 2]$ admissible by both,

$$\Delta_{\mathrm{ILPG}}(y) \leq \sqrt{2\tau} \leq \Delta_{\mathrm{Base}}(y) \qquad \text{(in expectation or with high probability)}. \tag{A.1}$$

**Theorem 6** (Robustness radius dominance). Under Assumptions 2–3 and equation A.1, consider prototype drift to the adapted prototypes. If the instance-specific drift satisfies

$$\max \left\{ \Delta_{\mathrm{ILPG}}(y), \max_{j \neq y} \Delta_{\mathrm{ILPG}}(j) \right\} \leq r,$$

then ILPG preserves the correct label provided $r < \gamma_{\mathrm{ILPG}}(x)/(2L)$. Similarly, the baseline requires $r < \gamma_{\mathrm{Base}}(x)/(2L)$. Consequently, if $\gamma_{\mathrm{ILPG}}(x) \geq \gamma_{\mathrm{Base}}(x)$ holds on average (or with high probability), then ILPG admits a *no-smaller* robustness radius:

$$r_{\mathrm{safe}}^{\mathrm{ILPG}} \geq r_{\mathrm{safe}}^{\mathrm{Base}}, \qquad r_{\mathrm{safe}}^{(\cdot)} := \frac{\gamma_{(\cdot)}(x)}{2L}.$$

*Proof.* For any competitor $j \neq y$, by Assumption 2,

$$g_y(\tilde{E}, S_W) - g_j(\tilde{E}, S_W) \geq g_y(\tilde{E}, S) - g_j(\tilde{E}, S) - L\big(\Delta(y) + \Delta(j)\big).$$

At zero drift this gap equals the margin in Assumption 3. Thus the sign of each pairwise gap is preserved if $L\big(\Delta(y) + \Delta(j)\big) < \gamma(x)$. A sufficient condition is $\max\{\Delta(y), \Delta(j)\} \leq r$ with $2Lr < \gamma(x)$, yielding the stated thresholds for ILPG and baseline. The dominance claim follows by comparing thresholds and using equation A.1. □

**Remark 2.** The constant $L$ can be upper-bounded from the network's local Lipschitz estimates (e.g., product of spectral norms of linear layers and Lipschitz constants of nonlinearities), and $\gamma$ can be approximated on a validation set.

**Generalization under drift.** Let $w_{y,u} = \langle S(y), S(u) \rangle$ and reuse the SAS notation. Suppose both methods share the same hypothesis class $F$, weights $\{w_{y,u}\}$, and samples, hence comparable complexity terms. Under the calibration coupling (Thm. A.2), and if ILPG achieves a no-larger weighted empirical seen loss, $\sum_y w_{y,u} \widehat{\mathbb{E}}_{D_y} L_y^{\mathrm{ILPG}} \leq \sum_y w_{y,u} \widehat{\mathbb{E}}_{D_y} L_y^{\mathrm{Base}}$, we obtain:

**Theorem 7** (Unseen risk dominance under drift). For any unseen class $u$ and confidence $\delta \in (0, 1)$,

$$\mathbb{E}_{D_u} L_u^{\mathrm{ILPG}} \leq \mathbb{E}_{D_u} L_u^{\mathrm{Base}} \quad \text{up to the same-order complexity/confidence terms,}$$

and the domain-shift term satisfies

$$\sum_y w_{y,u} \, \mathrm{TV}(D_u, D_y) \leq c \sum_y w_{y,u} \, \|S(u) - S(y)\|_2,$$

which is unaffected by replacing a static $S$ with a tighter (on-average) ILPG-adapted $S_W$ anchored by VSR.

*Proof.* Apply Theorem A.2 to both methods with identical $F, \{w_{y,u}\}, \{n_y\}$. The empirical loss dominance carries through the convex combination, complexity terms match up to constants. The TV-to-geometry reduction uses the same $c$, and VSR keeps $S_W$ near $S$, not enlarging the geometry term in expectation. □

**Consequence (operational form).** Combining Theorems 6 and 7: *under equal negative-similarity budget $\alpha$, comparable hypothesis complexity, and lower PCR/VSR/weighted-seen losses, ILPG has a no-smaller robust radius against prototype drift and a no-looser unseen-risk bound under semantic shift, hence is **provably superior or equal** to the baseline in these regimes.*

## C SUPPLEMENTARY VISUALIZATIONS

To provide a more comprehensive demonstration of ILPG's attribute localization capability, we include four supplementary attention-weight for unseen classes visualizations in the appendix Figure 8. The results reveal that the model can pinpoint target attributes with high precision, even when these attributes are partially occluded by irrelevant objects, the localization remains accurate and robust.

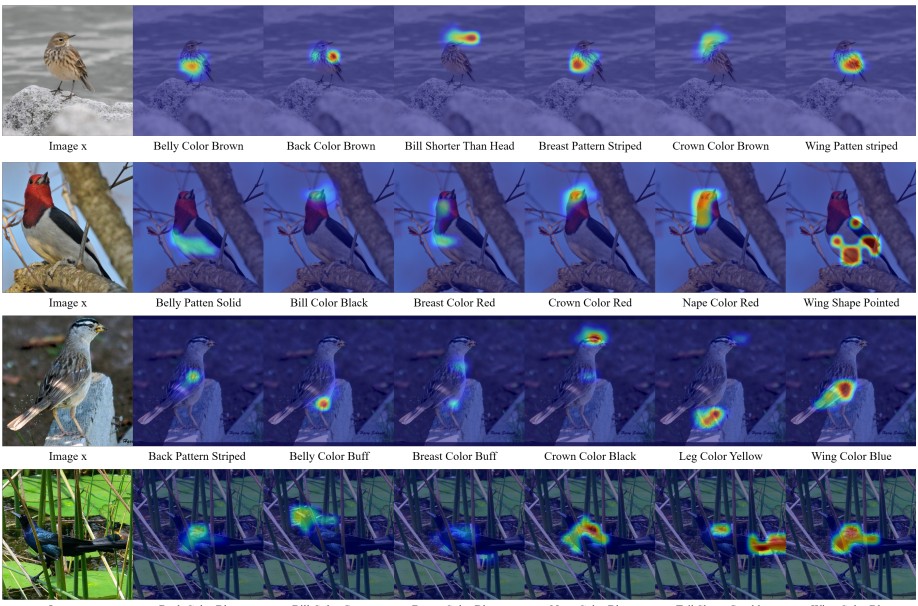

Figure 8: Visualization of attention maps for the ILPG.

## D SUPPLEMENTARY ABLATION STUDY

We present a supplementary ablation study of ILPG on the AWA2 dataset, with results summarized in Table 5. Each component consistently improves upon the Baseline, and the complete ILPG achieves the best overall performance. Specifically, the Baseline exhibits a pronounced bias toward seen classes in the GZSL task, yielding a poor H score. The SAS loss mitigates this bias by re-weighting predictions according to inter-class similarity, thereby boosting accuracy on unseen classes while preserving correct inter-class relationships.

## E CONFUSION MATRIX

Figure 9 presents the normalized confusion matrix (scaled to [0,1]) for the model on the CUB dataset. The main diagonal indicates correct predictions, with darker colors signifying higher accuracy. Non-diagonal elements reveal class confusion. Among the 200 classes, 0–49 are unseen, and 50–199 are seen. The ILPG model shows a clear main diagonal, particularly in the 0–49 range, with only a few lighter spots, highlighting its strong performance in CZSL. However, several bright spots in the 50–199 range indicate significant confusion between some unseen and seen classes. Future work will concentrate on enhancing the ability of model to distinguish between similar seen and unseen class features in the GZSL scenario to reduce cross-domain misclassification.

Table 5: Ablation results (%) of CZSL and GZSL on the AWA2 dataset.

| Method | GZSL | | | CZSL |
|---|---|---|---|---|
| | U | S | H | acc |
| Baseline | 7.0 | 94.8 | 13.0 | 65.4 |
| ILPG w/o SAS | 16.8 | 91.9 | 28.4 | 67.5 |
| ILPG w/o VSR | 55.0 | 82.1 | 65.9 | 65.8 |
| ILPG w/o PCR | 55.0 | 88.3 | 67.7 | 67.0 |
| ILPG w/o CSA | 50.3 | 90.0 | 65.5 | 67.0 |
| ILPG | **64.3** | **86.1** | **73.6** | **71.7** |

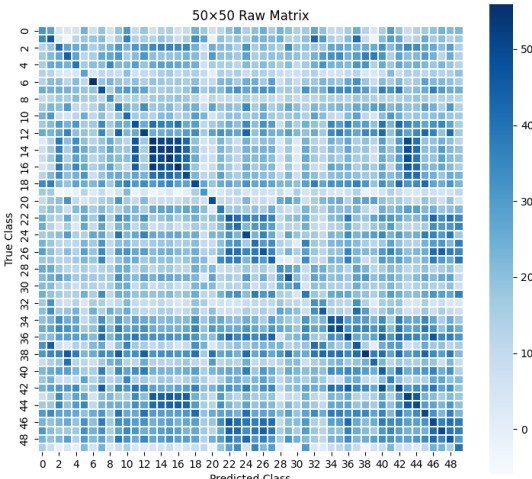

Figure 9: Visualization of Confusion Matrix for the ILPG.

## F   OCCLUSION ROBUSTNESS EXPERIMENTS

We evaluate ILPG and ILPG-IDPP under different occlusion ratios (3%, 5%, 7%, 10%, 20%, and 30%) on the CUB dataset. The quantitative results are summarized in Figure 10, showing the values of losses, unseen/seen accuracy, harmonic mean (H), and zero-shot accuracy (acc_zs).

## G   SUPPLEMENTARY HYPERPARAMETER ANALYSIS

As illustrated in Figures 12, we provide supplementary hyper-parameter analyses for ILPG on the SUN and AWA2 datasets. It is evident that an excessively small $\lambda_{SAS}$ leads to a noticeable drop in GZSL performance, whereas moderate values maintain strong and stable results. $\lambda_{VSR}$ and $\lambda_{PCR}$ exhibit some fluctuations

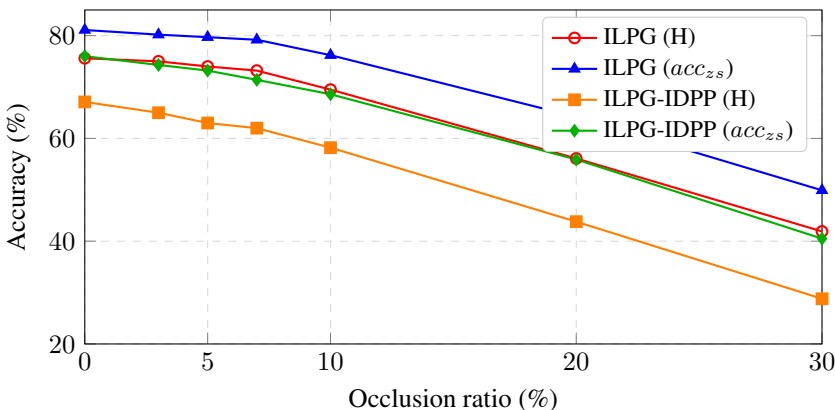

Figure 10: Occlusion robustness of ILPG vs. ILPG-IDPP on CUB. ILPG consistently shows smaller performance degradation under medium to heavy occlusion.

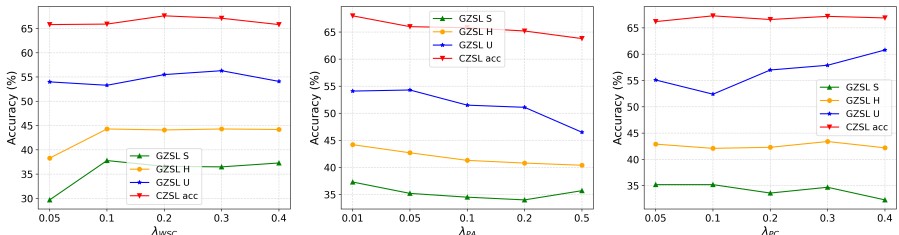

Figure 11: Hyperparameter analysis of $\lambda_{SAS}$, $\lambda_{VSR}$ and $\lambda_{PCR}$. We show the CZSL and GZSL performance variations on SUN.

within their search ranges, yet the overall performance remains robust across a wide span of choices. For SUN, we set the loss weights to $\lambda_{SAS} = 0.3$, $\lambda_{VSR} = 0.01$, $\lambda_{PCR} = 0.4$, for AWA2, we use $\lambda_{SAS} = 0.4$, $\lambda_{VSR} = 0.005$, $\lambda_{PCR} = 0.01$.

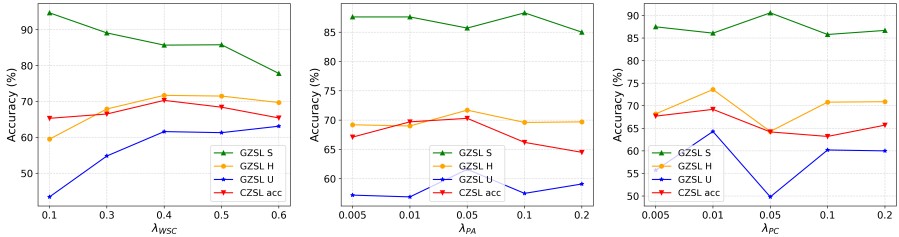

Figure 12: Hyperparameter analysis of $\lambda_{SAS}$, $\lambda_{VSR}$ and $\lambda_{PCR}$. We show the CZSL and GZSL performance variations on AWA2.

## H  THE USE OF LLMS

During the writing of this paper, we used a large language model (e.g., ChatGPT) to aid writing and text polishing. All research ideas, experiments, and results were independently completed by the authors.

## I  ALGORITHM

---
**Algorithm 1** Conventional Prototype-Based ZSL
---

**Input:** Training set $\mathcal{D}_s$, encoder $f(\cdot)$, class-level prototypes $S(c)$
Train encoder $f(\cdot)$ to map images into embedding space
**for** test image $x$ **do**
    Extract embedding $e = f(x)$
    Compute similarity scores with all prototypes:

$$P(y = c|x) = \frac{\exp\langle e, S(c)\rangle}{\sum_{j \in C} \exp\langle e, S(j)\rangle}$$

    Predict $\hat{y} = \arg\max_c P(y = c|x)$
**end for**
**return** predicted class label

---

---
**Algorithm 2** ILPG Training Procedure
---

**Input:** Training set $\mathcal{D}_s$, pre-trained encoder $f(\cdot)$, attribute embeddings $A$
**while** not converged **do**
    Sample batch $(x, y) \sim \mathcal{D}_s$
    Extract visual features $e = f(x)$
    Encode attributes into vectors $A$ (e.g., GloVe)
    Apply **AVL**: cross-attention between $e$ and $A$ to obtain localized features $\tilde{E}$
    Obtain instance signature $g$ from frozen DINOv2 encoder
    Apply **IDPP**: generate personalized prototype $S_W = W \odot S + S$ with $W = h(g)$
    Apply **APC**: match $\tilde{E}$ with $S_W$ to obtain class distribution $P(y|x)$
    Compute total loss:
$$L = L_{\text{IPM}} + \lambda_{\text{SAS}} L_{\text{SAS}} + \lambda_{\text{VSR}} L_{\text{VSR}} + \lambda_{\text{PCR}} L_{\text{PCR}}$$

    Update model parameters using gradient descent
**end while**
**return** trained ILPG model

---

---
**Algorithm 3** ILPG Inference Procedure
---

**Input:** Test sample $x$, trained ILPG model
Extract visual features $e = f(x)$
Encode attribute embeddings $A$
Apply **AVL** to obtain attribute-localized features $\tilde{E}$
Obtain instance signature $g$ from frozen DINOv2 encoder
Apply **IDPP** to generate personalized prototype $S_W$
Apply **APC** to compute class distribution $P(y|x)$
**return** predicted class label $\hat{y} = \arg\max_c P(y = c|x)$

---