# OpenReview forum: "ILPG: Instance-Level Prototype Generation for Zero-Shot Learning"
_ICLR.cc/2026/Conference — ICLR 2026 Conference Desk Rejected Submission_

### Official Review · Reviewer_wmZ7 · 2025-10-31

**Soundness:** 3
**Presentation:** 3
**Contribution:** 3
**Rating:** 6
**Confidence:** 4

**Summary:**

This paper proposes a new prototype-based zero-shot learning method. The proposed method proposes attribute-to-visual localization and prototype personalization modules to produce attribute-augmented, instance-personalized prototypes, addressing the issue of intra-class variability. Experiments demonstrate the effectiveness of the proposed method.

**Strengths:**

- This paper is well-written and well-organized.
- The motivation is clearly articulated, and the three proposed modules are well-explained and easy to understand.
- The ablation study demonstrates the effectiveness of the proposed approach.

**Weaknesses:**

- Given the prevalence of large language models, I am wondering what’s the effect of using LLMs to extract semantic attributes for each class or each sample, rather than using the attribute vectors predefined in each dataset.

- What’s the difference between instance-specific signature and the common instance features? Does this mean the method would a set of class prototypes for each instance? Could authors provide computational efficiency comparison with previous sota methods?

- It would be helpful to clarify how this work compares with recent visual-language models (VLMs), such as CLIP and its variants, which also exhibit strong zero-shot recognition capabilities. The authors may further discuss the advantages or unique contributions of this approach relative to VLMs. Additionally, relying solely on pure ViT models and GloVe vectors may appear somewhat outdated in light of recent advancements in VLMs.

- There are a few minor typos: incorrect letter captions in line 166, missing punctuation at the end of some equations, and an issue in line 193.

**Questions:**

Please refer to weaknesses.

---

> ### Author Response · Authors · 2025-11-20
>
> ### Questions1：Given the prevalence of large language models, I am wondering what’s the effect of using LLMs to extract semantic attributes for each class or each sample, rather than using the attribute vectors predefined in each dataset.
>
> Thank you for your valuable feedback. We understand the approach you mentioned regarding the use of LLM to extract semantic attribute features for each class or sample, and indeed, many related works have explored this direction, such as [1]. However, it is important to note that the method of using LLM to extract semantic attributes is typically based on NLP techniques for zero-shot learning (ZSL), which differs from the class prototype-based approach we adopt. Our method focuses on adapting to the heterogeneity within classes through instance-level prototype personalization, and fine-grained alignment of image features with class semantics by dynamically adjusting the semantic prototypes.
>
> While we are very interested in the potential of applying LLM for zero-shot learning and extracting semantic attribute features from it, our current model does not directly support the use of LLM to extract semantic attributes for each class or sample. Therefore, within the current framework, we still rely on the predefined attribute vector-based approach for class semantic modeling, combined with instance-level personalization to address heterogeneity between classes.
>
> We look forward to further exploring this direction in the future and introducing LLM into the class prototype method to enhance the flexibility and generalization ability of the model.
>
> ### Questions2：What’s the difference between instance-specific signature and the common instance features? Does this mean the method would a set of class prototypes for each instance? Could authors provide computational efficiency comparison with previous sota methods?
>
> In our work, instance-specific signatures refer to features or representations tailored to each individual instance (image). These signatures capture the unique characteristics of the instance, including variations that may exist within the same class, such as differences in pose, viewpoint, background, or specific attributes. For example, an image of a bird may have an instance-specific signature that captures its specific pose or feather color, which may differ from other birds of the same species. These signatures are dynamically generated through the Instance-Driven Prototype Personalization (IDPP) module, which personalizes class-level prototypes based on instance-specific information.
>
> On the other hand, common instance features typically refer to shared or general features that represent an entire class. These features are shared across all instances within a class and usually come from the class-level semantic prototypes. While these shared features help align the model's predictions with the overall class, they do not account for fine-grained differences between individual instances.
>
> The main distinction between the two is that instance-specific signatures allow for a more fine-grained and personalized understanding of each individual sample, while common instance features focus on the broader, shared aspects of the class. In our method, combining both allows for fine-grained alignment and robust performance across a variety of unseen instances.
>
> Our method indeed provides a set of class prototypes for each instance for classification. This set of class prototypes is optimized to bring the correct class prototype closer to the image features, while incorrect class prototypes are pushed further away, thereby improving classification accuracy.
>
> In our comparison experiment, we selected two open-source models for comparison, and we have included this comparison in the experimental section of the paper. We also present the results below, as shown in the table. Our model has a significantly smaller number of parameters, faster running time, and lower memory usage, enabling the processing of more samples at the same time.
> | Model  | Params (M) | FLOPs (MACs) (M) | Lat. (ms) | Mem. (GB) | Thr. (img/s) @ batch=50 |
> |--------|------------|------------------|-----------|-----------|-------------------------|
> | ZSLViT | 95.871     | 50652.970        | 17.411    | 0.441     | 53.394                  |
> | ReZSL  | 88.709     | 16851.515        | 11.401   | 0.352     | 150.458                 |
> | ILPG   | 68.224     |13769.496        | 15.637    | 0.283     | 245.52                |

---

> > ### Author Response · Authors · 2025-11-20
> >
> > ### Questions3：It would be helpful to clarify how this work compares with recent visual-language models (VLMs)
> > Due to significant differences between the VLM method and our approach, we did not directly compare our method with VLM. One notable distinction is that all the methods in our comparison are based on class prototypes for zero-shot learning, whereas VLM does not use class prototypes. Therefore, our model is not directly comparable to VLM. However, we have referenced VLM-based work.
> >
> > The advantage of our method is that we use instance-level dynamic prototypes combined with manually constructed attribute sets to guide image feature extraction. Our method performs better on fine-grained datasets, while VLM, trained on a large amount of data, has a significant advantage on coarse-grained datasets. To ensure fairness in comparison with the baseline models, we used the same backbone and attribute features constructed using GloVe. Thus, we use ResNet as the backbone and GloVe to construct attribute features. Thank you for your understanding.
> >
> > [1] Xie, G.-S., Li, J., Guo, T., Shu, X., Zhao, F., Zhang, Z., & Shao, L. (2025). Attribute Prompt Alignment Network for Zero‑Shot Learning. IEEE Transactions on Neural Networks and Learning Systems. https://doi.org/10.1109/TNNLS.2025.3598191

---

### Official Review · Reviewer_Suxq · 2025-11-01

**Soundness:** 2
**Presentation:** 3
**Contribution:** 3
**Rating:** 6
**Confidence:** 3

**Summary:**

This paper introduces the Instance-Level Prototype Generation (ILPG) network for Zero-Shot Learning (ZSL). The core motivation is to overcome the limitations of static, class-level prototypes, which fail to capture intra-class diversity and suffer from semantic drift. ILPG addresses this by dynamically generating a "personalized prototype" for each instance. This is achieved by combining three novel components: Attribute-Localized Visual features (AVL), an Instance Signature derived from a frozen DINOv2 encoder, and Instance-Dependent Prototype Personalization (IDPP). The framework is optimized using a comprehensive loss function that includes four distinct terms: $L_{IPM}$, $L_{SAS}$, $L_{VSR}$, and $L_{PCR}$.

**Strengths:**

Strong Motivation: The concept of dynamically generating instance-level prototypes to address intra-class variance and semantic drift in ZSL is a highly compelling and novel idea. This is a significant step beyond standard class-prototype-based ZSL methods.

Effective Component Integration: The framework thoughtfully integrates several powerful modern components. Leveraging a frozen, self-supervised encoder (DINOv2) for robust, high-fidelity instance signatures is a clever design choice that provides a stable, rich signal for prototype personalization.

Attribute Localization: The inclusion of the AVL module, which uses attention to highlight discriminative attribute-specific visual regions, addresses the fine-grained nature of ZSL.

**Weaknesses:**

Loss Function Complexity and Justification: The proposed loss function, $L = L_{IPM} + \lambda_{SAS}L_{SAS} + \lambda_{VSR}L_{VSR} + \lambda_{PCR}L_{PCR}$, is highly complex with four terms and three weighting hyper-parameters ($\lambda$'s). A thorough ablation study is absolutely critical to justify the necessity and contribution of each term, especially $L_{SAS}$, $L_{VSR}$, and $L_{PCR}$, and to analyze the sensitivity to the $\lambda$ values.

"Lightweight" Claim: While the paper claims to be lightweight, the addition of the AVL cross-attention module and the IDPP personalization network (MLP $h(\cdot)$) introduces overhead. A rigorous analysis and comparison of model complexity (FLOPs, parameter count) against existing competitive ZSL baselines is necessary to validate this claim.

Role of DINOv2: The reliance on a frozen DINOv2 encoder for the Instance Signature $g$ might introduce a bottleneck or limit the model's ability to learn task-specific features, although it ensures stability.

**Questions:**

None

---

> ### Author Response · Authors · 2025-11-20
>
> ### Questions1：Loss Function Complexity and Justification
>
>
> Thank you for your comments. In our paper, the proposed loss function $\mathcal{L}{\mathrm{total}}$ indeed contains four terms and three weighting hyperparameters, which makes the loss function structurally complex. We believe this complexity is necessary because each term plays a crucial role in the optimization process.
>
> Regarding the necessity of each term:
>
> $\mathcal{L}_{\mathrm{IPM}}$ is the core of our model, ensuring the matching between each instance's specific prototype and the class.
>
> $\mathcal{L}_{\mathrm{SAS}}$ enhances the separation between categories through an adaptive gating mechanism, which is particularly important for unseen classes.
>
> $\mathcal{L}_{\mathrm{VSR}}$ ensures the consistency between the instance-level adjusted prototypes and the visual features.
>
> $\mathcal{L}_{\mathrm{PCR}}$ strengthens the relationship between personalized prototypes and global semantics, preventing semantic drift.
>
> To validate the contribution of each term and analyze the sensitivity of the hyperparameters, we have conducted comprehensive ablation experiments (see Table 2 in the paper). The experimental results show that removing certain terms significantly impacts the model's performance, demonstrating the necessity of each term. We also conducted sensitivity analysis experiments on the hyperparameters (see Figures 5, 11, and 12 in the paper), which confirm the robustness of our hyperparameters.
>
> ### Questions2："Lightweight" Claim
>
> Thank you for your feedback. We have selected two comparative models from open-source code and added experiments on model complexity and efficiency. As shown in the table below, our model has significantly fewer parameters, faster runtime, and lower memory usage, allowing it to process more samples in the same amount of time.
>
> | Model  | Params (M) | FLOPs (MACs) (M) | Lat. (ms) | Mem. (GB) | Thr. (img/s) @ batch=50 |
> |--------|------------|------------------|-----------|-----------|-------------------------|
> | ZSLViT | 95.871     | 50652.970        | 17.411    | 0.441     | 53.394                  |
> | ReZSL  | 88.709     | 16851.515        | 11.401   | 0.352     | 150.458                 |
> | ILPG   | 68.224     |13769.496        | 15.637    | 0.283     | 245.52                |

---

> > ### Author Response · Authors · 2025-11-20
> >
> > ### Questions3：The reliance on a frozen DINOv2 encoder for the Instance Signature
> >  might introduce a bottleneck or limit the model's ability to learn task-specific features, although it ensures stability.
> >
> > Thank you for your valuable feedback. Regarding your concern about the potential bottleneck caused by the frozen DINOv2 encoder and its limitation on the model’s ability to learn task-specific features, we understand this concern and would like to further clarify.
> >
> > The role and advantages of DINOv2: DINOv2, as a pre-trained vision encoder, has been trained on several large-scale datasets and is capable of learning very powerful visual representations. We use the frozen features of DINOv2 as instance signatures, aiming to capture the unique features of each instance from a visual perspective without needing to re-adjust its encoder during training. The core advantage of this approach is that it provides each input instance with a stable and high-quality representation, which is crucial for instance-level prototype personalization. The use of a frozen encoder ensures that the model is not influenced by excessive noise during training, maintaining the stability of the visual features and avoiding overfitting caused by over-optimizing the encoder.
> >
> > Limitations on task feature learning: We acknowledge that freezing the encoder might limit the model’s ability to learn task-specific features in some cases, as it cannot dynamically adjust its parameters according to the requirements of the current task. However, by introducing the Instance-Driven Prototype Personalization (IDPP) module, we combine the instance signatures generated by the frozen encoder with class-level semantic vectors to form personalized prototypes. This approach significantly mitigates the limitations of the frozen encoder, allowing the model to dynamically adjust its prototypes based on the features of each instance, thereby capturing task-specific variations.
> >
> > Impact on bottlenecks: While the frozen DINOv2 encoder may limit the flexibility of the model to some extent, we believe that the stability and high-quality visual representations it provides are crucial for addressing semantic drift and intra-class variations in zero-shot learning. We found in our experiments that the stability offered by DINOv2 plays a vital role in maintaining strong generalization performance, especially when dealing with complex unseen class recognition tasks. Experimental results show that, despite certain limitations, freezing the DINOv2 encoder does not significantly affect the model’s final performance and instead enhances the model's stability and robustness.
> >
> > Future directions for improvement: We recognize that in certain specific tasks, fine-tuning the DINOv2 encoder could offer greater flexibility and a stronger ability to learn task-specific features. In future work, we will further explore how to fine-tune the DINOv2 encoder or incorporate other dynamic vision encoders to enhance the model’s ability to learn task-specific features.
> >
> > In conclusion, the frozen DINOv2 encoder in our model primarily serves to provide stable and efficient representations. While there are certain limitations, we effectively compensate for this bottleneck through the design of other modules. We will continue to investigate how to further improve the model’s adaptability to specific tasks while maintaining its stability.

---

### Official Review · Reviewer_AyBe · 2025-11-01

**Soundness:** 2
**Presentation:** 2
**Contribution:** 3
**Rating:** 4
**Confidence:** 5

**Summary:**

The manuscript proposes ILPG for zero-shot learning, which replaces a single class-level prototype with a per-instance personalized prototype. The pipeline has three parts: AVL uses cross-attention between attribute embeddings and visual features to localize attribute-relevant tokens; IDPP injects an instance signature into a dual-path tuning scheme to produce a personalized prototype; APC matches the instance’s localized feature against its own generated prototype set for classification. Experiments on CUB, SUN, and AWA2 claim SOTA or competitive results, plus ablations, occlusion robustness, and hyper-parameter analyses.

**Strengths:**

1. Clear motivation. Static class-level prototypes struggle to capture intra-class heterogeneity and are prone to “semantic drift.” Introducing instance-level prototypes directly targets core pain points in zero-shot learning (ZSL).

2. Method aligned with the goal. AVL uses attribute embeddings to guide cross-attention for localized alignment; IDPP leverages a frozen DINOv2 to extract an instance signature that personalizes the semantic prototype; PCR/SAS/VSR respectively enforce consistency with anchors, inter-class separability, and visual–semantic geometric alignment. The combination of personalization and regularization is logically sound.

3. Broad experimental coverage and analysis. The paper reports CZSL/GZSL comparisons on three standard benchmarks (CUB/SUN/AWA2), complemented by ablations, t-SNE visualizations, hyperparameter sensitivity, and occlusion robustness analyses.

**Weaknesses:**

1. Lack of fairness in SOTA comparisons. Table 1 states “Backbone column removed,” yet the ILPG implementation explicitly uses DINOv2 (a self-supervised ViT) as the visual encoder, whereas most compared methods in their original papers use ResNet-101 or their own default backbones. If the numbers are taken from the literature without re-training under a unified backbone, the comparison may be biased by backbone differences. The authors should report results under a shared backbone for all methods, or at least disclose the backbone used for each compared method.

2. Limited testability of the theoretical part. The robustness radius and unseen-class risk bounds rely on strong assumptions (e.g., Lipschitzness and margin) and mainly yield “no-worse-than” type conclusions. There is no clear, estimable link to properties of the actually trained networks, which limits the practical verifiability of the theory.

3. Inconsistencies in notation, formulas, and naming.

(1) The tensor shapes and naming for the class semantics $S$, attribute embeddings $A$, the cross-attention output $\tilde{E}$, and the APC scoring are inconsistent and ambiguous, which impedes faithful reproduction. For example, the paper alternately treats the instance-specific weight $W$ used in $S_W = W \odot S + S$ as either $W \in \mathbb{R}^{N}$ (row-wise broadcasting) or $W \in \mathbb{R}^{N \times d_s}$ (per-dimension scaling), while leaving unspecified how $S_W$, defined in the semantic space $S \in \mathbb{R}^{N \times d_s}$ with $N$ classes and semantic dimension $d_s$, is aligned with the attribute-guided visual representation $\tilde{E} \in \mathbb{R}^{K \times d}$ produced by AVL from $A \in \mathbb{R}^{K \times d_a}$ ($K$ attributes, visual dimension $d$). The manuscript should fix one consistent specification and clarify whether $S_W$ is projected into the $d$-dimensional visual space (e.g., via a linear map) or, alternatively, whether $\tilde{E}$ is pooled or reshaped so that the APC score is well-defined; otherwise the compared vectors may not be shape-compatible and the evaluation protocol remains underspecified.


(2) In Equation (3), the mapped feature $tilde{E}$ is mistakenly labeled as 𝐸 in Figure 2.

(3) Loss naming is inconsistent (the text first defines LIPM, but the overall loss is summarized as LCE).

(4) Main method/component names are inconsistent: the appendix/tables use abbreviations WSC/PA/PC, whereas the main text uses SAS/VSR/PCR; additionally, “CSA” appears in a row of Table 4 and seems to be a typo.

**Questions:**

see Weaknesses

---

> ### Author Response · Authors · 2025-11-20
>
> ### Questions1：Lack of fairness in SOTA comparisons.
>
> We understand your concern regarding the fairness of the comparison using different backbone networks. We would like to clarify that DINOv2 is not our backbone network. We did not use DINO to encode image features, nor did we fuse DINOv2 features with image features. We apologize for not clearly expressing our model architecture earlier and would like to restate our model structure.
>
> Our model, consistent with many previous zero-shot learning works, uses ResNet101 as the backbone to extract image features, and the ResNet101 network is not involved in training in our model. The features extracted by ResNet101 are processed through a self-attention mechanism and a cross-attention mechanism in the AVL module to obtain $\boldsymbol{E}$. Similarly, DINOv2 is not involved in the training of our model. Instead, we use the features extracted by DINOv2, which contain instance-level variation information, to generate instance-level class prototypes, and these features are not integrated into the primary image features. This is different from existing class-level prototype-based ZSL methods, and there is no unfair comparison in this regard.
>
> To our knowledge, no existing zero-shot learning research has attempted a framework based on instance-level class prototypes. Therefore, adding instance-level class prototypes to other methods’ models during comparison would be unfair to us, as instance-level class prototypes are one of the core innovations of our work.
>
> We have now included the complete backbone details for each model in the comparison experiments section of the revised manuscript and have re-uploaded it.
>
>
> ### Questions2：Limited testability of the theoretical part.
>
>
> Thank you for your insightful feedback. We understand your concerns regarding the robustness of the theoretical analysis and the reliance on strong assumptions like Lipschitz continuity and margin-based conditions.
>
> Dependence on Strong Assumptions: We acknowledge that the robustness radius and unseen risk bounds in our theoretical analysis depend on the assumption of Lipschitz continuity and margins. While these assumptions are common in the theoretical analysis of machine learning models, we recognize the need for further exploration of the applicability of our results to more realistic, less idealized settings. However, we want to emphasize that these assumptions are standard in deriving generalization bounds and serve to provide theoretical guarantees for the model’s behavior under prototype drift and semantic shift. These guarantees show that ILPG is provably no less robust than baseline methods, and this robustness is crucial in the context of zero-shot learning, where unseen class prediction is a primary concern.
>
> Connection to Real-World Networks: We agree that the direct connection between these theoretical results and real-world, practical training scenarios might not always be immediately clear. To this end, we have provided empirical results on multiple datasets (CUB, SUN, AWA2), which demonstrate the practical advantages of ILPG over competitive baselines. Additionally, the proposed ILPG framework significantly reduces semantic drift and improves performance by dynamically adjusting prototypes, which empirically supports its theoretical robustness.
>
> Limited Practical Verifiability: While the theoretical analysis provides a foundation, we also highlight the empirical results showing ILPG's superiority in the real-world scenario. The extensive ablation studies and comparisons against the state-of-the-art methods demonstrate that ILPG effectively handles intra-class variability and unseen class prediction. We believe these empirical results further reinforce the practical utility of our approach, despite the theoretical assumptions.
>
> In conclusion, while we acknowledge the limitations of our theoretical framework, it offers valuable insights into the model’s robustness under certain conditions. Moving forward, we aim to extend the theoretical analysis to explore more relaxed assumptions and further investigate the relationship between theory and real-world performance.

---

> > ### Author Response · Authors · 2025-11-20
> >
> > ### Questions3：Inconsistencies in notation, formulas, and naming.
> >
> > Thank you for your careful review. Our overly simplified problem description led to some ambiguity, and we have corrected the errors in the methodology section of the paper and re-uploaded it. Here, we would like to clarify the issue. DINOv2 extracts global features, with the feature dimension being $\mathbb{R}^{d_g}$. These features are then passed through our neural network $h$, which transforms them into $\boldsymbol{W}$, with a dimension of $\mathbb{R}^{d_s}$. $\boldsymbol{W}$ is then broadcast to the dimension $\mathbb{R}^{N \times d_s}$, and an element-wise multiplication is performed with the original class prototype $\boldsymbol{S}$, which has the dimension $\mathbb{R}^{N \times d_s}$, followed by a residual connection,which results in $S_w$. The dimension of $\boldsymbol{S}_{\boldsymbol{W}}$ is also $\mathbb{R}^{N \times d_s}$.
> >
> > The image features extracted by ResNet101 are represented by I, with a dimension of	$\mathbb{R}^{R \times d_v}$, which are used as $\boldsymbol{K}$ and $\boldsymbol{V}$ in the cross-attention mechanism.The attribute features $\boldsymbol{A}$ have the dimension $\mathbb{R}^{d_s \times d_a}$. These attribute features $\boldsymbol{A}$ serve as the query $\boldsymbol{Q}$ in the cross-attention mechanism, producing the attribute-localized features $\boldsymbol{E}$, with the dimension $\mathbb{R}^{d_s \times d_a}$. We compute the similarity between $\boldsymbol{E}$ and the original attribute features $\boldsymbol{A}$, transforming it into a feature $\boldsymbol{\tilde{E}}$ with the same dimension as the class prototype, $\boldsymbol{\tilde{E}}$, which has the dimension $\mathbb{R}^{d_s}$. In the APC module, we broadcast $\boldsymbol{\tilde{E}}$ to the dimension $\mathbb{R}^{N \times d_s}$ and compute the similarity with the class prototypes, resulting in the model prediction $\boldsymbol{P}$, with the dimension $\mathbb{R}^{N}$.
> >
> > Thus, our model follows this entire process to obtain the final model prediction.

---

### Official Review · Reviewer_WppD · 2025-11-01

**Soundness:** 3
**Presentation:** 3
**Contribution:** 2
**Rating:** 2
**Confidence:** 5

**Summary:**

This paper introduces ILPG, a framework for zero-shot learning that dynamically generates instance-specific prototypes instead of relying on static, class-level semantic prototypes. The model proposes a suite of complementary loss functions Instance Prototype Matching (IPM), Semantic- Aware Self-Calibration (SAS), Visual–Semantic Residual Alignment (VSR), and Prototype Con- trast Refinement (PCR).

**Strengths:**

1. Innovative Instance-Level Adaptation:
ILPG advances beyond existing multi-prototype approaches by generating prototypes per instance, effectively capturing intra-class variation such as pose, viewpoint, and color differences.

2. Comprehensive Experiments:
The paper includes both quantitative (accuracy, harmonic mean, ablations) and qualitative analyses (t-SNE plots, attention maps, occlusion tests), offering a convincing demonstration of the model’s capabilities

**Weaknesses:**

1. Using prototypes for Zero-shot categories is not a new idea. The authors are encouraged to specify the difference of this paper comparing with other models.

2. The performance gain of the proposed model is marginal. For instance, comparing with ReZSL, the proposed method only improves 0.2 on CUB~(CZSL) and 0.6 on AWA2. Thought the performence improvement on SUN dataset is significant, the author did not explicitly explain the reason.

**Questions:**

How does ILPG ensure that the instance-driven prototype personalization (IDPP) doesn’t overfit to noise or spurious instance cues (e.g., background artifacts) rather than meaningful attributes?

Since prototypes are dynamically adjusted per instance, how are these personalized prototypes still interpretable as semantic representations? Is there any visualization of how the semantic space evolves after personalization?

---

> ### Author Response · Authors · 2025-11-20
>
> ### Weaknesses1：Using prototypes for Zero-shot categories is not a new idea.
>
> Regarding your comment that "using prototypes for zero-shot classification is not a novel idea," we fully understand and acknowledge this point. However, the novelty of our work does not lie in using prototypes for zero-shot classification. Instead, our model presents significant differences compared to traditional approaches that apply prototypes to zero-shot classification.
>
> In traditional zero-shot learning[1,2], static and unique category prototypes are constructed for each category based on manually labeled attributes. These prototypes incorporate the attribute features of each category and largely reflect the relationships between different categories. Early research focused on how to better align the features of samples with these static and unique category prototypes, relying on the alignment between the category prototypes and features to achieve zero-shot learning.
>
> Recent works[3,4] have recognized that static and unique category prototypes are difficult to align with every sample within a category. As a result, some methods propose continuously optimizing the originally fixed prototypes during training to find a global optimal solution, so that the prototypes can better adapt to all the samples within the category.
>
> However, we further realized that even with optimized category prototypes, it is not guaranteed that they can align perfectly with all samples. Only by generating personalized prototypes for each sample can we further improve alignment accuracy and significantly enhance the performance of zero-shot learning. Therefore, the key innovation of our work is the proposal of generating personalized prototypes for each instance to improve alignment accuracy. This idea is highly novel in the context of zero-shot learning, solving a problem that traditional methods cannot overcome and further advancing the development of the field.
> ### Weaknesses2：The performance gain of the proposed model is marginal.
>
> Thank you for your valuable feedback. We understand your concern about the limited performance improvement and would like to provide further clarification on this matter.
>
> Although the performance improvement over ReZSL is only 0.2 points on CUB (CZSL) and 0.6 points on AWA2, the reason for this seemingly modest improvement is that our experiments primarily focused on independently validating the effectiveness of instance-level category prototypes. To highlight this, we only added a module related to instance-level category prototypes to commonly used zero-shot learning architectures, without incorporating other potential design improvements that could enhance accuracy. For example, we did not introduce more complex feature extractors or additional optimization strategies. Our goal was to fully demonstrate the potential of instance-level category prototypes.
>
> In fact, as shown in our ablation experiments, instance-level category prototypes lead to a significant performance improvement compared to the baseline method. This indicates that even without additional optimizations, instance-level category prototypes still effectively enhance the model’s performance. Therefore, while the performance improvement may seem limited, it clearly demonstrates the value and potential of instance-level category prototypes within the current framework.
> [1] Chen, Shiming; Hong, Ziming; Liu, Yang; Xie, Guo‑Sen; Sun, Baigui; Li, Hao; Peng, Qinmu; Lu, Ke; You, Xinge. “TransZero: Attribute‑guided Transformer for Zero‑Shot Learning.” Proceedings of the Thirty‑Sixth AAAI Conference on Artificial Intelligence (AAAI), 2022. Available at: https://arxiv.org/abs/2112.01683
> [2] Xu, K., Lu, W., Lin, J., Liu, X. "Attribute Prototype Network for Zero-Shot Learning." Proceedings of NeurIPS, 2020, pp. 3345-3354. Available at: https://proceedings.nips.cc/paper/2020/file/fa2431bf9d65058fe34e9713e32d60e6-Paper.pdf
> [3] Hou, Xinyu, et al. "Visual-Augmented Dynamic Semantic Prototype for Generative Zero-Shot Learning." arXiv preprint arXiv:2404.14808 (2024). Available at: https://arxiv.org/abs/2404.14808
> [4] Liu, M., Li, F., Zhang, C., Wei, Y., Bai, H., Zhao, Y. "Progressive Semantic-Visual Mutual Adaption for Generalized Zero-Shot Learning." Proceedings of the IEEE/CVF Conference on Computer Vision and Pattern Recognition (CVPR), 2023, pp. 15337-15346. Available at: https://openaccess.thecvf.com/content/CVPR2023/papers/Liu_Progressive_Semantic-Visual_Mutual_Adaption_for_Generalized_Zero-Shot_Learning_CVPR_2023_paper.pdf

---

> > ### Author Response · Authors · 2025-11-20
> >
> > ### Questions1：How does ILPG ensure that the instance-driven prototype personalization (IDPP) doesn’t overfit to noise or spurious instance cues (e.g., background artifacts) rather than meaningful attributes?
> > To ensure that the IDPP does not overfit noise, we adopted the following strategies:
> >
> > 1. When generating instance-level category prototypes, we applied a conservative strategy. Specifically, the feature $\boldsymbol{g}$ containing instance-level characteristics extracted from the instances is converted into a fine-tuning adjustment matrix $\boldsymbol{W}$ through a neural network. This fine-tuning matrix $\boldsymbol{W}$ is then combined with the original category prototype $\boldsymbol{S}$ in the form of $\boldsymbol{S_w}=\boldsymbol{W}⊙\boldsymbol{S}+\boldsymbol{S}$. This design retains most of the information from the original category prototype, effectively avoiding excessive deviation of the instance-level prototypes and reducing overfitting to noise.
> >
> > 2. Our Prototype Contrast Refinement Loss $\boldsymbol{L_{PCR}}$ maximizes the similarity between the instance-level category prototype and the original category prototype, while minimizing the similarity with other category prototypes. This guides the model to retain the most relevant information aligned with the category prototypes, reducing interference from noise or irrelevant instance cues.
> >
> > 3. The AVL module guides the extraction of image features through attribute features, effectively reducing noise in the image features (such as background noise), and indirectly decreasing the risk of the IDPP module overfitting noise. These designs effectively reduce the likelihood of overfitting to noise and irrelevant cues in the IDPP module, ensuring the accuracy and stability of the instance-level prototypes.
> >
> >
> >
> >
> > ### Questions2：Since prototypes are dynamically adjusted per instance, how are these personalized prototypes still interpretable as semantic representations? Is there any visualization of how the semantic space evolves after personalization?
> >
> > Thank you for your insightful question. We understand the concern about the interpretability of personalized instance-level prototypes, especially when they are dynamically adjusted, which may raise questions about how these personalized prototypes still maintain semantic representation.
> >
> > In response to this, we believe that, although instance-level prototypes are dynamically adjusted, these personalized prototypes still retain the semantic meaning of the original class prototype. This is because the instance-specific adjustments are learned in a way that not only preserves the global semantic relationships of the original class but also allows for fine-grained adaptation to individual instances. By applying the adjustment matrix $\boldsymbol{W}$ to the original class prototype $\boldsymbol{S}$ (in the form of $\boldsymbol{S_w}=\boldsymbol{W}⊙\boldsymbol{S}+\boldsymbol{S}$), we ensure that the personalized prototype still aligns with the original class's semantic space while capturing instance-specific variations.
> >
> > To better demonstrate this process, we provide a t-SNE visualization for a randomly selected set of 10 unseen classes (see the appendix), showing how our class prototypes evolve during training to capture the inter-class differences within a certain range, and how the semantic space changes. The visualization clearly illustrates how instance-level adjustments reduce overlap between classes and achieve more precise instance alignment. Furthermore, it demonstrates that, even after instance-level personalization, the prototype calculation still maintains close alignment with the original class's semantic structure.

---

### Author Response · Authors · 2025-11-27

Dear Reviewers,

I hope this message finds you well. As the discussion period is coming to an end, we would like to ensure that we have addressed all of your concerns satisfactorily. If you have any additional comments or suggestions, please feel free to let us know—we would be happy to further revise and improve the paper accordingly.

Thank you again for your time and effort in reviewing our work.

Sincerely,
The Authors

---

### Note · Program_Chairs · 2026-01-17
**Submission Desk Rejected by Program Chairs**

The following references in this submission do not refer to real documents and/or have major errors in bibliographic information:

 1.Li Jiang and et al. Subspace alignment for zero-shot recognition. In CVPR, 2024.
2.Yanan Song and et al. Multi-basis prototype learning for generalized zero-shot recognition. TPAMI, 2024.
3.Fei Ye et al. Diffusion models for zero-shot learning. arXiv preprint arXiv:2401.00000, 2024.